# Formation of Lipid-Derived Flavors in Dry-Cured Mackerel (*Scomberomorus niphonius*) via Simulation of Autoxidation and Lipoxygenase-Induced Fatty Acid Oxidation

**DOI:** 10.3390/foods12132504

**Published:** 2023-06-27

**Authors:** Qiaoyu Liu, Menglin Lei, Wenhong Zhao, Xiangluan Li, Xiaofang Zeng, Weidong Bai

**Affiliations:** 1College of Light Industry and Food Sciences, Academy of Contemporary Agricultural Engineering Innovations, Zhongkai University of Agriculture and Engineering, Guangzhou 510225, China; urnotmu@163.com (M.L.); zhaowengong2002@126.com (W.Z.); leexiangluan@163.com (X.L.); xiaofang_zeng2015@163.com (X.Z.); weidong_bai2010@163.com (W.B.); 2Guangdong Key Laboratory of Lingnan Specialty Food Science and Technology, Zhongkai University of Agriculture and Engineering, Guangzhou 510225, China; 3Key Laboratory of Green Processing and Intelligent Manufacturing of Lingnan Specialty Food, Ministry of Agriculture, Guangzhou 510225, China

**Keywords:** lipoxygenase, fatty acids, autoxidation, enzymatic oxidation, volatile compounds

## Abstract

In this study, lipoxygenase (LOX) extracted from dry-cured mackerel was purified, resulting in a 4.1-fold purification factor with a specific activity of 493.60 U/min·g. LOX enzymatic properties were assessed, referring to its optimal storage time (1–2 days), temperature (30 °C), and pH value (7.0). The autoxidation and LOX-induced oxidation of palmitic acid (C16:0), stearic acid (C18:0), oleic acid (C18:2n9c), linoleic acid (C18:2n6c), arachidonic acid (C20:4), EPA (C20:5), and DHA (C22:6n3) were simulated to explore the main metabolic pathways of key flavors in dry-cured mackerel. The results showed that the highest LOX activity was observed when arachidonic acid was used as a substrate. Aldehydes obtained from LOX-treated C18:1n9c and C18:2n6c oxidation, which are important precursors of flavors, were the most abundant. The key flavors in dry-cured mackerel were found in the oxidative products of C16:0, C18:0, C18:1n9c, C18:2n6c, and C20:4. Heptanaldehyde could be produced from autoxidation or LOX-induced oxidation of C18:0 and C18:1n9c, while nonal could be produced from C18:1n9c and C18:2n6c oxidation. Metabolic pathway analysis revealed that C18:1n9c, C18:2n6c, EPA, and DHA made great contributions to the overall flavor of dry-cured mackerel. This study may provide a relevant theoretical basis for the scientific control of the overall taste and flavor of dry-cured mackerel and further standardize its production.

## 1. Introduction

Mackerel (*Scomberomorus niphonius*) is one of the most important commercial marine fishes in the world and is a high source of protein (≥15 g/100 g) [1], polyunsaturated fatty acids (e.g., docosahexaenoic acid, arachidonic acid, and linoleic acid), vitamins, and minerals [2].

Different processing methods confer mackerel with varying flavors. Zhang, Wang, Chu, Sun, and Lin [3] indicated that the thermal processing of mackerel produces undesirable mutagenic and/or carcinogenic agents (heterocyclic aromatic amines and advanced glycation end products). Dry curing with salt is a simple and representative method for processing raw mackerels, during which microorganisms are inhibited due to reduced moisture content and water activity. As one of the determinants of the unique sensory characteristics of dry-cured meat products, flavor plays an important role in evaluating the freshness, taste, and nutritional value of dry-cured meat products [4]. After salting in 3–6% salt and drying, the unique salty flavor of mackerel is formed due to the oxidative hydrolysis of lipids and proteins. Lipid oxidation has the greatest impact on the formation of mackerel flavor [5]. Li, Liu, Wang, Liu, and Peng [6] revealed that the key volatile compounds in salted silver carp, such as heptanal, octanal, 1-octene-3-ol, 1-pentene-3-ol, and 2-heptanone, were mainly generated by unsaturated fatty acids (USFAs). Wang, Wang, Wu, Xiang, Zhao, Chen, Qi, and Li [7] found that four small molecular substances—phenylacetaldehyde, cinnamaldehyde, 2, 4-sebacedienal, and 2-ethyl-1-hexanol—which significantly affected the flavor of the golden pomfret, were generated after dry-curing and the content of unsaturated fatty acids decreased significantly. This finding indicated that the flavor composition and content were closely associated with the type and content of fatty acids (FAs). Aldehydes, which are closely associated with fish flavor, can be produced by FAs, especially USFAs, through autoxidation and enzymatic oxidation. These may have varying effects on the formation of flavors.

Lipid oxidation continuously occurs during the processing of dry-cured mackerel, accumulating sufficient FAs that are further degraded into different hydroperoxides and then degraded into stable small-molecule compounds that play an important role in the overall flavors of dry-cured mackerel. USFAs may have a greater impact on flavor formation in dry-cured mackerel owing to its unsaturated bonds and increased breaking sites. An, Wen, Li, Zhang, Hu, and Xiong [8] indicated that the autoxidation of lipids has a huge contribution to the formation of aldehydes, alcohols, and phenols during the process of gel formation in surimi. Similarly, Han, Zhang, Li, Wang, Chen, and Kong [4] reported that the lipid oxidation involved in the process of cured meat production was mainly autoxidation. In enzymatic oxidation, free radical oxidation is mainly caused by lipoxygenase (LOX). LOX can trigger the generation of free radicals, accelerating the degradation of FAs to produce different types of hydroperoxides, followed by further oxidation to produce volatile or non-volatile compounds [9]. Zhang, Hua, Li, Kong, and Chen [10] discovered that the production of key flavors in peas and soybeans was relevant to the LOX and FA contents. Zhang, Pan, Zhou, Wang, Dang, He, Li, and Cao [11] studied aroma formation during the process of curing boneless ham, observing that LOX was a key enzyme in the oxidation of flavor precursor substances, which was beneficial to improving the quality of the ham.

LOX has hydrophilic groups and is a water-soluble protease. Currently, salt fractionation is commonly used to initially purify the crude enzyme, and both phosphate and sulfate can precipitate part of the protein. Moreover, ammonium sulfate is the most widely used reagent due to its low-temperature coefficient and high solubility, which can reduce damage to the active components of LOX. The high-resolution purification and identification of LOX include gel chromatography, ion exchange chromatography, and electrophoresis, of which ion-exchange chromatography can be used to separate proteins with different charges under acid–base conditions, while gel chromatography is used to separate proteins with different molecular weights.

However, until now, the dominant lipid oxidation pathway in aquatic organisms has not been reported. Therefore, in this study, LOX extracted from dry-cured mackerel was purified and parts of its properties were assessed. Additionally, the purified LOX was employed and the autoxidation and LOX-induced oxidation of palmitic acid (C16:0), stearic acid (C18:0), oleic acid (C18:1n-9), linoleic acid (C18:2n-6), arachidonic acid (C20:4n-6), EPA (C20:5n-3), and DHA (C22:6n-3) were simulated to explore the main metabolic pathways of key flavors generated by FAs in the dry-cured mackerel. This study may provide a relevant theoretical basis for the scientific control of the overall taste and flavor of dry-cured mackerel and further standardize its production.

## 2. Materials and Methods

### 2.1. Materials and Reagents

Chilled mackerel was purchased from Hailan Kitchen Co., Ltd. (Haikou, China). Methanol (chromatographically pure), chloroform (analytically pure), isopropanol (chromatographically pure), hexane (chromatographically pure), 4-methyl umbel ketone (>99%), phospholipid standards, linoleic acid standard (>99%), and 37 types of fatty acid methyl ester mixed standards were purchased from Sigma-Aldrich (Saint Louis, MO, USA). Phosphoric acid buffer (pH = 7.0) and citric acid (analytically pure) were bought from Guangzhou Chemical Reagent Factory (Guangzhou, China). N-alkanes (C6-C40) were purchased from o2si Biotechnology Co., Ltd. (Beijing, China). Acetic acid, ethyl ether, n-heptane, 2,4,6-trimethylpyridine (TMP), potassium hydroxide, Ethylene Glycol Tetra-acetic Acid (EGTA), dithiothreitol (DTT), ethylenediamine tetra-acetic acid (EDTA), and disodium hydrogen phosphate (Na2HPO4) were purchased from Aladdin Bio-Chem Technology Co., Ltd. (Shanghai, China).

### 2.2. Preparation of Dry-Cured Mackerel

Chilled mackerels were thawed at room temperature. The head, tail and viscera of the mackerels were removed. The fish were washed with ultrapure water, drained, and weighed. The middle part of the mackerel was cut into pieces (200 ± 1.0 g) 20 × 10 × 2 cm in size. Edible salt (4% of dry fish weight) was evenly spread on both sides of the fish. Thereafter, the mackerel pieces were placed in a 15 °C artificial climate chamber (BSP250, Boxun Medical Instrument Co., Ltd., Shanghai, China) to air dry for 5 d. Five sampling points were set up during the dry-cured processing of mackerel: fresh fish (A), end of curing (B), air-dried for 1 d (C), air-dried for 3 d (D), and air-dried for 5 d (E). At least 6 pieces of fish were taken from each sampling site. After removing fish skin and bones, the fish pieces were mixed and ground using a high-speed homogenizer (Jintan Liangyou Instrument Co., Ltd., Changzhou, China). The dry-cured mackerel was stored at −40 °C until further analysis.

### 2.3. Extraction of Crude Enzyme Solution and Determination of Lipidoxygenase (LOX) Activity

Extraction of the crude enzyme and determination of its activity was conducted as described in a previous study [12]. Crushed fish sample (15 g) was mixed thoroughly with 60 mL of 0.05 mol/L phosphoric acid buffer solution (pH = 7.0) containing 0.001 mol/L EGTA and DTT in ice water for 4 × 10 s at a speed of 8000× *g*. The homogenized mixture was then shaken for 30 min and centrifuged for 60 min at 10,000× *g* at 4 °C. The supernatant was collected. Linoleic acid (140 mg) was dissolved in 5.0 mL of distilled water and 180 μL of Tween 20 (0.0125%, *v*/*v*). The pH value of the mixture was adjusted to 9.0 using 2 mol/L NaOH solution and the mixture was then made up to 50 mL with distilled water. Citric acid buffer (0.05 mol/L, pH = 5.5) was mixed with 200 μL of linoleic acid substrate in a 20 °C water bath. The mixture was put into a spectrophotometer (U-2000, Hitachi, Tokyo, Japan) to observe its change at 234 nm (ε = 2.5 × 104 M^−1^ cm^−1^) at 25 °C. Once the absorbance value was stable, 0.1 mL of crude LOX solution was added. Changes in absorbance values after 1 min were calculated. One unit of LOX activity was defined as the amount of enzyme required to increase absorbency by 1 U/min and the result was expressed as U/min·g sample. The protein concentration was estimated using the Bradford method, with bovine serum albumin as the standard protein.

### 2.4. Isolation and Purification of LOX

Fractional precipitation using ammonium sulfate

All steps were conducted below 4 °C following a previously described method [13]. An appropriate amount of crude enzyme solution was mixed with 114 g/L anhydrous ammonium sulfate solution in an ice water bath with stirring at a constant speed for 20 min. The mixture was passed through a double-layer filter paper or gauze to collect the filtrate. The supernatant was mixed with 243 g/L anhydrous ammonium sulfate solution in an ice water bath with stirring at a constant speed for 20 min. After the homogenate was centrifuged at 30,000× *g* for 20 min, the supernatant was discarded and the residue collected and dissolved in an equal volume of phosphate buffer. Afterward, the mixture was recentrifuged at 90,000× *g* for 20 min to obtain the supernatant.

Dialysis

The enzyme solution was precipitated with 20–40% ammonium sulfate solution and then placed in dialysis bags and dialyzed with 50 mmol/L phosphate buffer (pH = 7.0) at 4 °C, during which the external solution was changed at an interval of 2–3 h. The supernatant was centrifuged and freeze-dried after overnight dialysis.

DEAE Sepharose Fast Flow (FF) Anion Exchange Chromatography

A DEAE Sepharose FF column was equilibrated with 50 mmol/L phosphate buffer (pH 7.0). The dialyzed enzyme solution was dissolved in an equal volume of phosphate buffer. The adsorbed protein was eluted from the column with a linear gradient of 0–0.5 mol/L NaCl at a rate of 0.8 mL/min. The absorption peak was measured at a wavelength of 280 nm and the tube with peak absorption was collected. Samples with LOX activity were collected, freeze-dried, and stored for further analysis.

Electrophoresis

Vertical plate sodium dodecyl sulfate−polyacrylamide gel electrophoresis (SDS−PAGE) was performed as previously described in the study by Aanangi, Kotapati, Palaka, Kedam, Kanika, and Ampasala [14]. After electrophoresis, the molecular weights of the samples were determined after staining with 0.25% CBB R-250 and decolorizing with glacial acetic acid-methanol.

### 2.5. Enzymatic Characteristics (Storage Stability, Optimal Temperature, pH Value, and Substrate Specificity of LOX Extracted from Dry-Cured Mackerel)

Storage stability

Enzymatic properties were measured as described in a previous study [15]. Briefly, the extracted LOX solution was stored at 25 ± 3 °C for 7 days and its enzyme activity was measured at the same time every day.

Optimal temperature and pH value for LOX activity in dry-cured mackerel

The extracted LOX solution was dissolved in separate phosphate buffer solutions (50 mmol/L) with pH values of 6.6, 7.0, 7.4, 7.8, 8.2, and 8.6 and then heated in water baths set at 10, 20, 30, 40, 50, and 60 °C for 20 min. Afterward, the extracted LOX solution was mixed with separate solutions of linoleic acid, linolenic acid, and arachidonic acid to determine its activity.

Substrate specificity

The extracted LOX solution was reacted with linoleic acid, linolenic acid, and arachidonic acid, and enzyme activity was determined according to the method described in 2.3.

### 2.6. Determination of Volatile Compounds in a Simulated Lipid Oxidation System

Simulation of lipid oxidation of FAs

A total of 15 µL of FA standards (C16:0, C18:0, C18:1n9c, C18:2n6c, C20:4, C20:5, and C22:6n3) were dissolved in 5 mL of phosphate buffer solution (0.05 mol/L PBS, pH = 7.0), 20 mg of extracted crude LOX solution, and 10 µL of Tween 80 in 15 mL bottles. The mixture was mixed for 15 s using a mixer and then stirred in a 30 °C water bath for 30 min [16].

Headspace Static Solid Phase Microextraction-Gas Chromatography-Mass Spectrometer (HS-SPME-GC-MS) analysis

HS-SPME conditions: The enzymatically oxidized and autoxidized samples (1.00 ± 0.01 g) were separately mixed with 0.2 μL of 2-octanol and the bottle was quickly closed. The activated 65-μm DVB-PDMS extractor was inserted and adsorbed on a magnetic stirring table at 30 °C for 40 min. After extraction, the samples were inserted into a GC/MS injection port for analysis. The analysis conditions were 250 °C for 10 min. GC conditions: The fractions were applied to a DB-5MS column (30 m × 0.25 mm × 0.25 μm) with a flow rate of 1.0 mL/min using helium gas at a constant linear rate. The split ratio was 1:20 and the injection volume was 1 μL. The initial temperature was held at 35 °C for 1 min, raised to 5 °C/min and kept at 60 °C for 1 min, then increased to 140 °C at 6 °C/min and held here for 1 min. Finally, the temperature was raised to 230 °C at 8 °C/min and held here for 5 min. MS conditions: Ion source temperature was 200 °C; ionization energy was 70 eV; mass scan range was between 35 and 350 *m*/*z* with no solvent excision time. Qualitative and quantitative analyses of volatile compounds: Agilent GCMS software version 9.0 was used to compare the retention time of volatile compounds with the spectrum library NIST17.L, and compounds with a matching degree > 80 were screened. The ratio of the peak area of volatile compounds to the peak area of the internal standard of known concentration was used to calculate the content of volatile compounds.

### 2.7. Data Analysis

At least 6 pieces of mackerel samples from each stage were used and the experiment was replicated three times. IBM SPSS Statistics (version 26, Chicago, IL, USA) was used for variance analysis and Duncan’s multiple range comparison. *p* < 0.05 represented significant differences. GraphPad Prism software (version 9.0.0, San Diego, CA, USA) was used for plotting. GC-MS data were processed using Agilent GCMSD software (Agilent Technologies, Inc., Santa Clara, CA, USA).

## 3. Results

### 3.1. Isolation and Purification of LOX in Dry-Cured Mackerel

The crude LOX solution extracted from dry-cured mackerel was purified via 20–40% ammonium sulfate fractional precipitation, dialysis, and DEAE Sepharose FF ion-exchange chromatography. The protein content and enzyme activity of LOX before and after purification were measured, and the purification fold was calculated by the LOX-specific activity. The results are shown in Table 1. The LOX-specific activity of crude LOX solution was 120.38 U/min.g, and the purity improved slightly after ammonium sulfate precipitation and dialysis. The dialyzed LOX solution was further purified using DEAE Sepharose FF ion-exchange chromatography, with its specific activity reaching 493.60 U/min.g, a multiplication of 4.10, indicating that LOX purity was highly increased.

Afterward, SDS-PAGE was performed on both crude and purified samples to further visually observe the efficiency of LOX purification (the results are presented in Figure 1). Based on the SDS-PAGE, the crude enzyme bands were more numerous and more blurred, suggesting that the crude enzyme contained more impurities. However, after ammonium sulfate precipitation, dialysis, and DEAE Sepharose FF ion-exchange chromatography, there was only one band in passage 2 at 95 kDa. As reported in previous studies, the molecular weight of LOX was approximately 95 kDa [17,18], which indicates that the LOX solution obtained in the final step was highly purified.

### 3.2. Enzymatic Characteristics of LOX Extracted from Dry-Cured Mackerel

#### 3.2.1. Storage Stability of LOX

The storage stability of LOX extracted from the dry-cured mackerel was measured at the same time over 7 consecutive days. The results are shown in Figure 2a. LOX-specific activity decreased by 97% with the prolongation of storage time (*p* < 0.05), from 85.23 (day 0) to 2.51 U/min.g (day 7). On the first day of storage, LOX activity was 80.57 U/min.g, representing 95% of the initial enzyme activity. LOX activity decreased sharply after two days of storage, and no significant activity was found on day 7. A previous study of LOX extracted from tarpon fish showed results similar to those of our study, i.e., that LOX activity remained at a high level in the first two days of storage but the enzyme began to deactivate and denature on day 3. The reason for this short shelf life of free LOX could be that after being separated from mackerel, there is no substrate and no suitable viable environment for LOX, ultimately leading to the deactivation and degeneration of the enzyme.

#### 3.2.2. Optimal Temperature for LOX Activity

The effects of temperature on the activity of LOX extracted from mackerel were measured using water baths with different temperatures. As shown in Figure 2b, the optimal temperature for LOX activity is 30 °C. LOX activity increased slowly between 10 and 30 °C, reaching 93% activity at the optimal temperature, indicating that LOX activity could be higher than that observed at lower temperatures. The increased temperature (10–30 °C) promoted LOX activation, enhancing LOX activity [19]. However, LOX activity decreased sharply when the temperature increased beyond 30 °C, with the activity decreasing to 54% at 40 °C and then to 8% of the optimal temperature at 60 °C, indicating complete inactivation. Banerjee, Khokhar, and Apenten [20] extracted LOX from carp muscle and found that it was inactivated at 65 °C because the tertiary structure of LOX changed irreversibly, thus inactivating the catalytic site of the enzyme [21]. Taken together, these results show that LOX in dry-cured mackerel could be a heat-sensitive enzyme.

#### 3.2.3. Optimal pH Value for LOX Activity

LOX activity in different pH buffer solutions was measured and the results are shown in Figure 2c. The optimal pH for LOX was 7.0. LOX activity at pH 6.6 was 59% of LOX activity at the optimal pH, indicating that the activity of LOX extracted from mackerel was relatively lower in acidic conditions. LOX activity began to decrease when the pH was higher than 7.0. When the pH was 8.8, LOX activity decreased to 22% of that observed at the optimal pH, indicating a loss of almost all enzyme activity. This was consistent with the results obtained by López-Nicolás, Pérez-Gilabert, and García-Carmona [22]. Under too acidic or too alkaline conditions, the protein structure is damaged, the polar groups connected to LOX are dissociated, and the center structure of the enzyme is irreversibly changed, thus inhibiting LOX activity. According to a previous report by Perraud, Kermasha, and Bisakowski [23], there may be two optimal pH values for LOX, which could be due to the LOX isoenzymes. In this study, only one optimal pH value was found, indicating that the LOX type extracted from mackerel was relatively simple and interference from multiple LOX synergies was excluded.

#### 3.2.4. Specificity of LOX Substrates

The optimal substrates for LOX extracted from fish may vary greatly. C18:2n6c, C18:3n3, and C20:4 are common optimal substrates for fish [24]. These three substrates were selected and reacted with LOX extracted from mackerel in this study. LOX activities are shown in Figure 2d. When C18:2n6c, C18:3n3, and C20:4 were used as substrates, LOX activities were 117.14, 79.30, and 123.57 U/min·g, respectively. It was obvious that C20:4 was the most suitable substrate for mackerel-derived LOX. LOX activity with C18:2n6c as substrate was 95% of that with C20:4 as substrate, suggesting that C18:2n6c could also be used as a suitable substrate for LOX. When C18:3n3 was used as a substrate, LOX activity was 65% that of C20:4. These observations suggest that mackerel-derived LOX can react with different FA substrates. Most of these USFAs are precursors of volatile compounds that can be catalyzed by LOX for flavor formation.

### 3.3. Profiles of Volatile Compounds Generated by Different FAs

FAs are among the most important flavor precursors. Different FA substrates can be degraded to produce different flavor substances. In this study, the degradation products of key FAs in dry-cured mackerel were investigated. The volatile compounds produced from reacting C16:0, C18:0, C18:1n9c, C18:2n6c, C20:4, C20:5, and C22:6n3 with mackerel-derived LOX are shown in Figure 3. Based on the number of volatile compounds, the highest number of volatile compounds (49) was obtained when C18:2n6c was used as the substrate, followed by C18:1n9c (where a total of 37 volatile compounds were detected). Aldehydes were detected when C16:0, C18:0, C18:1n9c, C18:2n6c, and C20:4 were used as substrates, with C18:1n9c and C18:2n6c generating the highest number of aldehydes. Alcohols were only detected when C18:1n9c was used as the substrate, while ketones were identified when C18:0, C18:1n9c, and C18:2n6c were used as substrates. Alkane olefine, aromatics, and esters were all detected from these seven types of FAs. Generally, volatile compounds were more abundant, and the number of aldehydes was highest, when C18:1n9c and C18:2n6c were used as substrates. Similarly, Zhang, Hua, Li, Kong, and Chen [10] and Sha, Lang, Sun, Su, Li, Zhang, Lei, Li, and Zhang [25] found that C18:1n9c and C18:2n6c had positive effects on flavor formation.

### 3.4. The Effects of LOX Treatment on Volatile Compounds Derived from FA Oxidation

#### 3.4.1. Changes in the Contents of Volatile Compounds Produced from C16:0 and LOX-Induced C18:0 Oxidation

The changes in the contents of volatile compounds produced from C16:0 and LOX-induced oxidation are reported in Table 2. The concentration of volatile compounds in LOX-treated C16:0 reached 135.75 μg/L, which was higher than without LOX, indicating that LOX promoted C16:0 oxidative degradation. In the absence of LOX, C16:0 was mainly degraded to octane, methyl tetracanoate, and methyl hexadecanoate, of which hydrocarbons were mainly generated by the autoxidation of alkyl radicals in the system [26]. Heptanal, nonal, 2-heptanone, and D-limonene were detected at concentrations of 1.32, 7.06, 12.42, and 0.54 μg/L, respectively, after LOX-induced C16:0 oxidation. Thus, it can be concluded that heptanal, nonal, 2-heptanone, and D-limonene can be produced by the enzymatic oxidation of C16:0. Heptyl aldehyde has a fishy, sweet almond smell. Nonyl aldehyde renders fat incense, fragrant grass, and fishy. 2-heptyl ketone and D-limonene are fruity. Collectively, it can be postulated that the enzymatic oxidation of C16:0 made a significant contribution to flavor formation. Moreover, C16:0 accumulated during the dry-curing process, providing sufficient substrate for further degradation.

#### 3.4.2. Changes in the Content of Volatile Compounds Produced from C18:0 and LOX-Induced C18:0 Oxidation

As indicated in Table 3, the total concentration of volatile compounds in LOX-treated C18:0 reached 289.71 μg/L, which was 0.6 times higher than without LOX, suggesting that LOX promoted the oxidative degradation of C18:0. On the other hand, the total concentration of volatile compounds in the autoxidation system was 172.40 μg/L. C18:0 could be degraded to produce aldehydes, ketones, alkanes, and esters, with heptane being present at the highest concentration (53.78 μg/L), indicating that strong alkyl radical reactions occurred during autoxidation. The types of aldehydes and ketones increased with LOX treatment. Nonal and 2-nonone were newly detected substances, at concentrations of 7.8 μg/L and 7.31 μg/L, respectively. LOX promoted C18:0 to produce nonal and 2-nonone. The concentration of heptanaldehyde was 20.73 μg/L under LOX treatment, which was 0.52 times higher than without LOX treatment (*p* < 0.05). This suggests that LOX improved the oxidative degradation of C18:0. Moreover, no significant change was observed in the concentration of 2-heptanone with or without LOX treatment (*p* < 0.05), indicating that 2-heptanone was mainly produced by autoxidation. These degradation products had a positive effect on the overall flavor of mackerel. For instance, heptanal provided a fishy and sweet apricot flavor, while nonal provided an oily and green grass flavor, further demonstrating the importance of C18:0 as a flavor precursor.

#### 3.4.3. Changes in the Contents of Volatile Compounds Produced from C18:1n9c and LOX-Induced C18:1n9c Oxidation

Table 4 displays the changes in the concentrations of volatile compounds produced from C18:1n9c oxidation with or without LOX. Overall, the concentration of volatile compounds by the autoxidative system was 1020.50 μg/L and increased by 0.53 times in the LOX-induced oxidative system, reaching 1563.68 μg/L. A total of 28 types of C18:1n9c oxidative products were detected, with esters accounting for the highest number (nine), followed by aldehydes (six). (E)-2-decenal, heptanal, decanal, hexanal, nonanal, and octanal were all detected in the two oxidation systems. (E)-2-decenal production in the presence of LOX treatment had no significant change compared with its production in the absence of LOX, while heptanal, Kuabal, hexal, nonal, and octanal production in the presence of LOX treatment showed an obvious increase compared to their production in the absence of LOX (*p* < 0.05). 1-nonanol, 2-dodecenol, heptanol, and octanol are the top four types of alcohols produced by the degradation of C18:1n9c. 2-dodecenol was detected after LOX treatment. Furthermore, 2-decanone, 2-nonanone, 2-undecanone, and 3-octanone are the top four ketones produced by the degradation of C18:1n9c. However, 3-octanone was detected only after LOX treatment. Most of the C18:1n9c degradation products can be produced by autoxidation and LOX enzymatic oxidation, with 2-undecanone, heptanal, kexal, hexanal, nonanal, and octanal being the key flavors in dry-cured mackerel. This further demonstrates that the oxidative products of C18:1n9c contribute significantly to the overall flavor of dry-cured mackerel.

#### 3.4.4. Changes in the Contents of Volatile Compounds Generated from C18:2n6c and LOX-Induced C18:2n6c Oxidation

The concentrations of volatile compounds in C18:2n6c and LOX-treated C18:2n6c oxidative systems were 2362.20 and 2469.81 μg/L, respectively (Table 5). A total of 49 types of C18:2n6c oxidative products were detected, including 6 aldehydes, 4 ketones, 10 alkenes, 4 aromatic hydrocarbons, 22 esters, and 3 other substances. (E,E)-2, 4-nonanodienal, heptanal, hexal, nonanoaldehyde, and hydrocarnital were detected in both oxidative systems, with their concentrations increasing with LOX treatment. Heptanaldehyde was detected only with LOX treatment, indicating that heptanaldehyde is mainly produced via LOX-induced oxidation. 2-heptanone, phenylbutanone, phenylpentanone, and thalidone are the top four ketones produced by the degradation of C18:2n6c. The concentration of 2-heptanone was 57.96 μg/L with LOX treatment, 0.31 times higher than without LOX treatment, suggesting that LOX promotes the formation of 2-heptanone from C18:2n6c. Heptanaldehyde, hexanal, and nonanal are key flavors in dry-cured mackerel.

#### 3.4.5. Changes in the Contents of Volatile Compounds Produced from C20:4 and LOX-Induced 20:4 Oxidation

The concentrations of volatile compounds produced by C20:4 treated with LOX was 2100.93 μg/L, which was 1.29 times higher than without LOX treatment, indicating that LOX enhanced the oxidative degradation of C20:4 (Table 6). A total of 26 types of oxidative products of C20:4 were detected, including 3 aldehydes, 8 alkenes, 9 aromatic hydrocarbons, 3 esters, and 3 other substances. The three aldehydes were trans,trans-2,4-decanodienal, hexal, and carlinal; trans,trans-2,4-decanodienal and carlinal were detected with LOX treatment, indicating that these two aldehydes were mainly produced by C20:4 via LOX-induced oxidation. The concentration of hexal after LOX treatment was 3.88 μg/L, 0.60 times higher than without LOX addition. The alkane olefines were significantly increased after LOX treatment (*p* < 0.05), suggesting that LOX promotes the oxidation of alkyl radicals. Furthermore, hexanal was the only volatile compound produced by C20:4 degradation that was a key flavor in dry-cured mackerel.

#### 3.4.6. Changes in the Contents of Volatile Compounds Produced from C20:5 and LOX-Induced C20:5 Oxidation

The total content of volatile compounds in the LOX-treated C20:5 group was 567.53 μg/L, which was 0.45 times higher than without LOX treatment (392.36 μg/L) (Table 7). The key oxidative products of C20:5 (aromatic hydrocarbons and alkenes, aldehydes, alcohols, and ketones) were not detected. However, six types of alkane olefines and five types of aromatics were detected after LOX treatment, which suggests that LOX may aggravate the oxidation of alkyl radicals. Similarly, the total contents of volatile compounds in the C22:6n3 and LOX-treated C22:6n3 groups were 1943.45 and 1642.75 μg/L, respectively (Table 8). A total of 31 types of oxidative products of C22:6n3 were detected, including 10 types of alkene olefines, 12 types of aromatics, 6 types of esters, and 3 types of other substances. The main olefines were decane and undecane, while the main aromatics were m-xylene and o-xylene. Due to their high thresholds [27], most olefines and aromatics contribute less to flavors. Toluene has a typical fish aroma, and most of the toluene from C20:5 were derived from LOX-induced oxidation, with a relatively low concentration of 15.45 μg/L, which may not contribute much to flavor formation. Therefore, volatile compounds derived from C20:5 and C22:6n3 may have no significant effect on the overall flavor of dry-cured mackerel.

### 3.5. Changes in Volatile Compounds Generated from Simulated FA Oxidation System

The concentrations of volatile compounds produced from oxidative degradation of the seven types of FAs were visualized using a heat map and the outcome is shown in Figure 4, where red represents a higher concentration of volatile compounds and green depicts lower concentrations. Overall, the products of oxidative degradation of different FAs differ greatly. In terms of the number of volatile compounds, the oxidative products of C18:1n9c and C18:2n6c are the most abundant, while those of C16:0 and C18:0 are the least abundant. This is because C16:0 and C18:0 are saturated FAs and do not contain unsaturated bonds, leading to relatively stable structures [28]. The concentrations of oxidative products of these FAs increased with LOX treatment. The increment was larger with LOX treatment when C18:1n9c and C20:4 were used as substrates. Therefore, it was inferred that C18:1n9c and C20:4 were suitable substrates for LOX. Herein, it was found that LOX activity was highest when C20:4 was used as a substrate, indicating that it was the optimal substrate for LOX. An and Oh [29] reported findings similar to those of this study. Although the total concentration of volatile compounds produced by LOX-treated C18:2n6c did not increase significantly, the concentration of the aldehydes, which was strongly associated with fish flavor, was 0.85 times higher than that produced via autoxidation, suggesting that LOX plays a vital role in promoting C18:2n6c as a flavor precursor. Ghorbani Gorji, Calingacion, Smyth, and Fitzgerald [30] also found that LOX plays a catalytic role in the C18:2n6c-related flavor of mayonnaise.

The oxidative products produced by C16:0, C18:0, C18:1n9c, C18:2n6c, and C20:4 formed key flavors of dry-cured mackerel. Compared with using C18:1n9c and C18:2n6c as substrates, less heptanaldehyde and nonaldehyde were produced when C16:0 and C18:0 were used as substrates. Heptanal, decal, hexal, nonal, octanal, and 2-undecanone are key flavors in C18:1n9c oxidative products that can be produced by autooxidation or LOX-induced oxidation. After LOX treatment, the concentrations of heptanal, decal, hexal, and 2-undecanone increased 0.15, 0.48, 0.40, 0.75, 0.16, and 0.20 times, respectively, with LOX having a greater effect on the production of nonal, consistent with most reports [31], indicating that C18:1n9c is an important flavor precursor. Heptanal, hexal, and nonal are products of C18:2n6c oxidation. The concentrations of hexal and nonal increased 0.63 and 0.71 times, respectively, with LOX treatment. This suggests that LOX had greater activity when C18:2n6c was converted to hexal and nonal. 13-hydroperoxylinoleic acid is cleaved between C12 and C13 to produce hexanal [28,32]. Heptanaldehyde was newly produced through LOX-induced oxidation, and it is postulated that LOX catalyzes C18:1n9c to form heptanaldehyde. However, its mechanism remains unclear. C20:4 was oxidized and degraded to hexal, and its concentrations from autoxidation and LOX-induced oxidation were 2.44 and 3.88 μg/L, respectively. This increase may be due to the increased content of hydroperoxides, promoting the conversion of C20:4 to hexaldehyde. As C20:5 and C22:6n3 contain more unsaturated bonds, their degree of oxidation is higher. Moreover, LOX can promote its oxidation [33,34]. Even though the products of C20:5 and C22:6n3 oxidation did not include the key flavors in dry-cured mackerel, they contained the characteristic flavors of fish, including toluene (fish aroma) and D-limonene (fruit aroma).

### 3.6. Metabolic Pathways of Lipid-Derived Flavors in Dry-Cured Mackerel

The metabolic pathways of C18:1n9c, C18:2n6c, C20:4, C18:3n3, C20:5, and C22:6n3 were mapped using information from the KEGG pathway, the Metacyc database, and a query of the literature, as shown in Figure 5a. The pathways of flavor formation are shown in Figure 5b. The contents of C18:1n9c, C18:2n6c, C20:4, C18:3n3, DHA, and EPA in mackerel increased continuously during the dry-curing process due to the continuous hydrolysis of fatty acyl CoA to unsaturated FAs catalyzed by acylCoA thioesterase. Linoleic acid could also be produced from lecithin through phospholipase A2 catalysis.

#### 3.6.1. C18:1n9c Metabolic Pathway

Nonal, octanal, decal, heptanal, and decanol are the main oxidative products of C18:1n9c, of which nonal, octanal, decanal, and heptanol are the key flavors in dry-cured mackerel. Nonal, octanal, and heptanol were produced by mackerel LOX-induced C18:1n9c oxidation, and a significant increase was observed compared with autoxidation (*p* < 0.05). This suggests that LOX-induced oxidation of C18:1n9c plays a bigger role compared with autoxidation. C18:1n9c mainly generates 10-hydroperoxide, 11-hydroperoxide, and 8-hydroperoxide through autoxidation and enzymatic oxidation. Under the catalysis of hydroperoxidase lyase (HPL), 10-hydroperoxide is degraded to nonal, 11-hydroperoxide is degraded to octanal, and 8-hydroperoxide is degraded to decal. It can be reduced to the corresponding alcohols by alcohol dehydrogenase (ADH) [35]. In conclusion, C18:1n9c is an important lipid-derived flavor precursor in dry-cured mackerel (Figure 5c).

#### 3.6.2. C18:2n6c Metabolic Pathway

C18:2n6c produces various hydroperoxides, including 10(R)-hydroperoxides, 9(S)-hydroperoxides, 13(S)-hydroperoxides, 8(R)-hydroperoxides, and 11(S)-hydroperoxides) through autoxidation and enzymatic oxidation. A variety of volatile compounds (e.g., trans-2-octene-1-ol, 1-octene-3-ol, 2, 4-decendienal, hexal, and heptanol) are produced under the catalysis of HPL, of which 1-octene-3-ol, hexal, heptanal, and heptanol are the key flavors in dry-cured mackerel. The mackerel LOX-induced C18:2n6c oxidation also produces hexal and heptanal. C18:2n6c is catalyzed by linoleate 10R-lipoxygenase and LOX to form 10(R)-hydroperoxides, forming 1-octene-3 alcohol by HPL. It can also be formed via a free radical reaction. 13(S)-hydroperoxide is generated under the catalysis of LOX2S and LOX15, and is then degraded to acetaldehyde and heptanaldehyde by HPL, and further degraded into hexanol and heptanol by ADH. Acetaldehyde can also be generated through autoxidation (Figure 5c).

#### 3.6.3. C20:4 Metabolic Pathway

C20:4 is catalyzed by arachidonic acid-lipoxygenase (ALOX) to produce various hydroperoxide eicosanotetraenoic acids such as 8(S)-HPETE, 12(R)-HPETE, and 12(S)-HPETE, as well as 12-hydroperoxides. Hydroperoxide eicosapenoic acid is further degraded into hydroxyl eicosapenoic acid (8(S)-HETE, 12(R)-HETE, and 12(S)-HETE), while 12-hydroperoxides are degraded into a variety of volatile compounds. 1-pentene-3-alcohols and (E) 2-octene-1-alcohols are generated from the degradation of 12-hydroperoxides by HPL [36], accounting for 6.30–15.34% and 2.77–4.64% of the volatile alcohols in dry-cured mackerel, respectively. 1-pentene-3-alcohol has a green and fishy aroma, while (E) -2-octene-1-alcohol has a fat aroma. Therefore, C20:4 oxidative products contribute to the overall flavor of dry-cured mackerel (Figure 5d).

#### 3.6.4. C18:3n3 Metabolic Pathway

C18:3n3 is mainly catalyzed by LOX2S to form (9S,10E,12Z,15Z)-9-hydroperoxy-10,12,15-octadecarbotrienoate (9(S)-HPETE) and (9Z,11E,15Z)-(13S)-hydroperoxide octadecane-9,11,15-trioleate (13(S)-HPOT), which can be degraded into 3,6-nonadienal and 3-hexanal by HPL. 3-hexanal is reduced to 3-hexanol under the catalysis of ADH and then further degraded into (Z)-3-hexene-1-ol acetate by 3-hexene-1-ol acetyl transferase (CHAT). Although C18:3n3 accounts for a certain proportion (2.16–2.74%) of the FAs in dry-cured mackerel, its oxidative products were not detected in the volatile compounds of dry-cured mackerel based on the metabolic pathways. This indicates that C18:3n3 may not influence the overall flavor of dry-cured mackerel (Figure 5c).

#### 3.6.5. EPA and DHA Metabolic Pathways

C20:5 (EPA) and C22:6n3 (DHA) are important long-chain ω-3 polyunsaturated fatty acids (PUFAs) in fish, with DHA being the most abundant FA in dry-cured mackerel. The flavors of dry-cured mackerel can be detected in the oxidative products of LOX-treated EPA and DHA or through their autoxidation. 1-pentene-3-alcohol is the oxidative product of EPA, while heptanaldehyde is the oxidative product of DHA. EPA generates 12-hydroperoxides in the presence of LOX and is then cleaved to 1-pentene-3-ol by HPL [37]. In addition, DHA can be degraded into 13-hydroperoxide; β-cleavage of a C-C single bond at both ends of the hydroperoxyl group can then produce heptanaldehyde [38]. Therefore, EPA and DHA may play positive roles in the generation of the overall flavor of dry-cured mackerel (Figure 5c).

## 4. Conclusions

In conclusion, the key flavors of dry-cured mackerel were found in the oxidative products of C16:0, C18:0, C18:1n9c, C18:2n6c, and C20:4. Moreover, C18:1n9c, C18:2n6c, EPA, and DHA contributed greatly to the overall flavors of dry-cured mackerel. Dry-cured mackerel has a complex enzyme system, and additional enzymes such as hydroperoxidase and LOX isoenzyme can be explored in future studies to further understand the mechanism of lipid-derived flavor formation in aquatic products. This study may provide a relevant theoretical basis for scientific control of the overall taste and flavor of dry-cured mackerel and further standardize its production.

## Figures and Tables

**Figure 1 foods-12-02504-f001:**
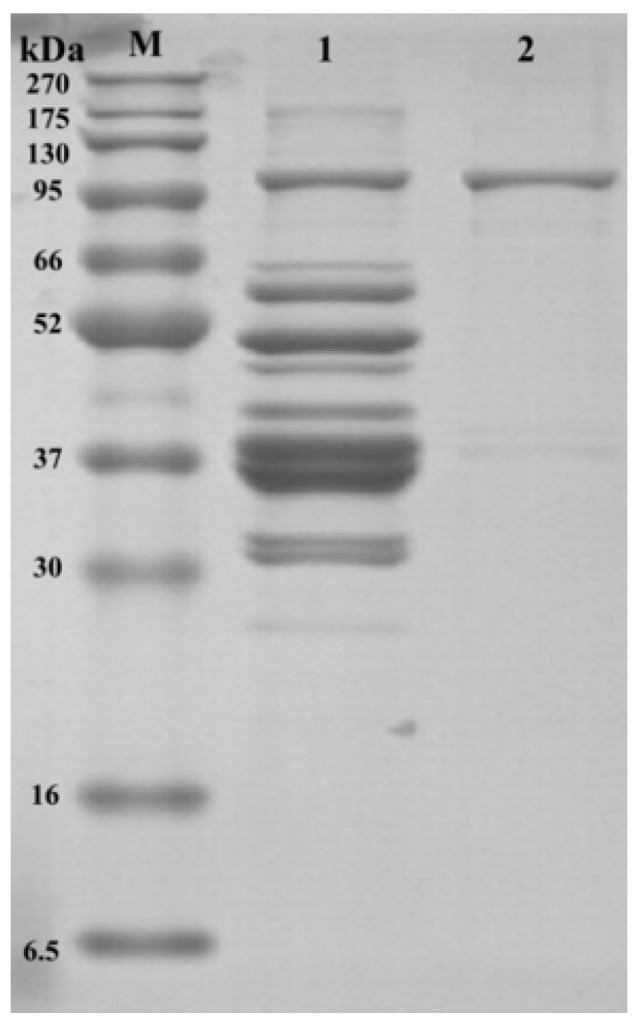
SDS-PAGE of purified and crude LOX in dry-cured mackerel. M = protein marker; Band 1: crude LOX solution; Band 2: purified LOX solution.

**Figure 2 foods-12-02504-f002:**
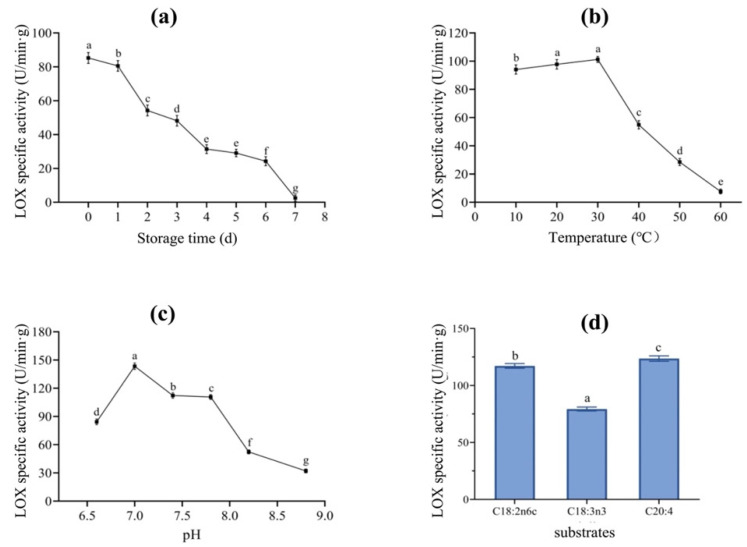
Enzymatic characteristics of LOX extracted from dry-cured mackerel at different storage times (**a**), temperatures (**b**), pH values (**c**), and substrates (**d**). Values are mean ± standard deviation (SD). Values in different points (**a**–**c**) or in different bars (**d**) with lowercase letters represent significant difference between each other.

**Figure 3 foods-12-02504-f003:**
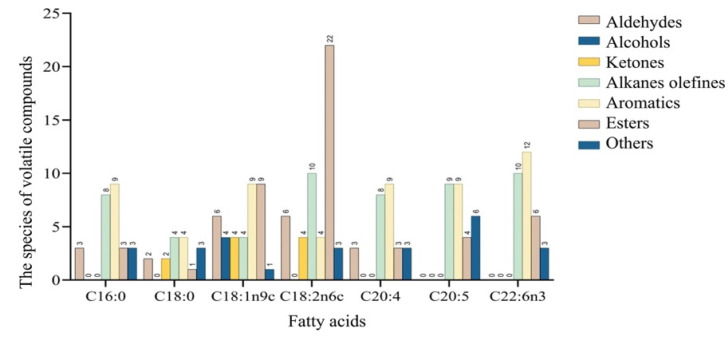
The number of volatile compounds produced by fatty acid oxidation.

**Figure 4 foods-12-02504-f004:**
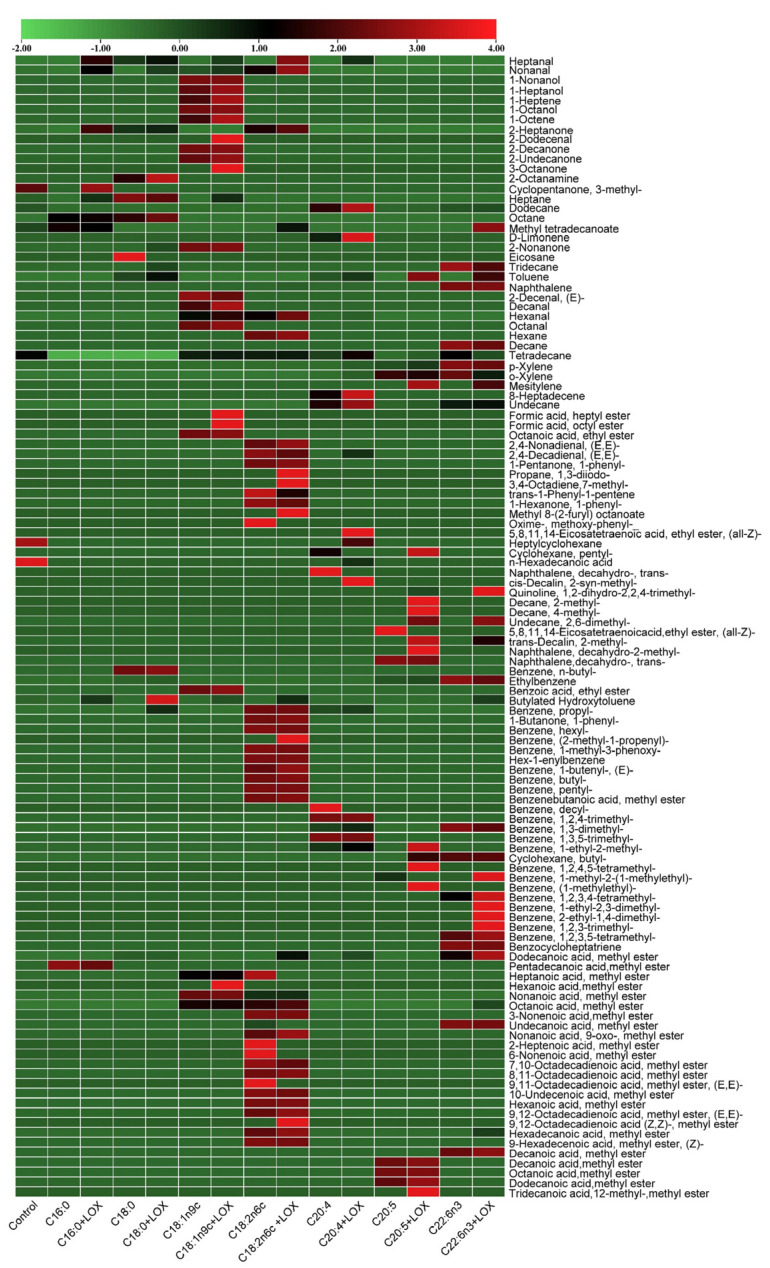
Heat map of the oxidation products of seven fatty acids.

**Figure 5 foods-12-02504-f005:**
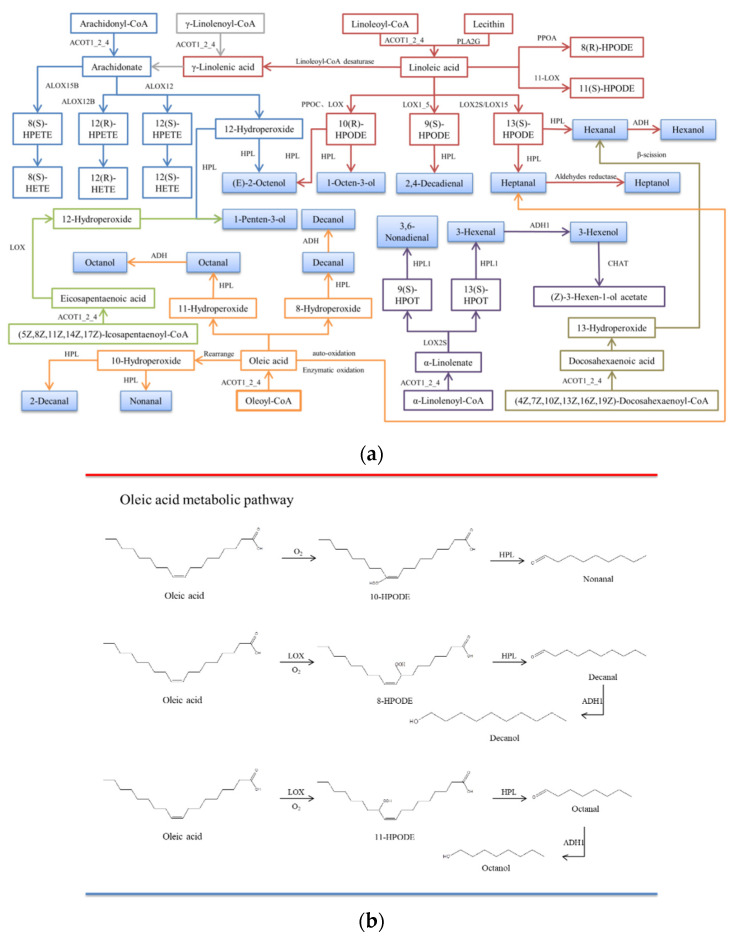
(**a**) The main metabolic pathways of unsaturated fatty acids. CoA: Coenzyme A; ACOT1_2_4: Acyl-CoA thioesterase; PLA2G: secretory phospholipase A2; PPOA: Linoleate 8R-lipoxygenase; PPOC: Linoleate 10R-lipoxygenase; LOX: lipoxygenase; LOX1_5: linoleate 9S-lipoxygenase; ALOX15B: 8-lipoxygenase (S-type); ALOX12: Arachidonic acid 12-lipoxygenase; ALOX12B: arachidonic acid 12-lipoxygenase (R type); HPL: hydrogen peroxide lyase; ADH: alcohol dehydrogenase class P; CHAT: 3-hexene-1-ol acetyltransferase; 9(S)-HPOT: (9S,10E,12Z,15Z)-9-hydroperoxy-10,12,15-octadecarbotrienoate; 13(S)-hpot: (9Z,11E,15Z)-(13S)-hydroperoxide octadecane-9,11,15-trioleate; HPETE: hydrogen peroxide eicosaptaenoic acid; HETE: hydroxyl eicosaptaenoic acid; HPODE: hydrogen peroxide lipid. (**b**) Oleic acid metabolic pathway. (**c**) Linoleic acid and arachidonic acid metabolic pathways. (**d**) Linolenic acid, EPA, and DHA metabolic pathways.

**Table 1 foods-12-02504-t001:** Purification of LOX in dry-cured mackerel.

Purification	Protein Content (g/L)	Activity (U)	LOX Activity (U/min.g)	Purification Fold
Crude enzymes	0.91	109.33	120.38	1.00
Fractional precipitation with ammonium sulfate	1.25	171.67	137.07	1.14
Dialysis	1.62	231.67	142.92	1.19
DEAE Sepharose FF	0.23	114.67	493.60	4.10

**Table 2 foods-12-02504-t002:** Changes in the concentrations of volatile compounds produced from C16:0.

Number	Compounds	Concentration of Volatile Compounds (μg/L)		
		Blank	C16:0	C16:0 + LOX
1	Heptanal	0.00 ± 0.00 a	0.00 ± 0.00 a	1.32 ± 0.49 b
2	Nonanal	0.00 ± 0.00 a	0.00 ± 0.00 a	7.06 ± 0.95 b
3	2-(6-ethoxy-4-methylquinazoline -2-ylamino)5-methylpyrimidine-4,6-diol	0.00 ± 0.00 a	0.00 ± 0.00 a	1.32 ± 0.14 b
4	2-Heptanone	0.00 ± 0.00 a	0.00 ± 0.00 a	12.42 ± 1.55 b
5	Cyclopentanone, 3-methyl-	0.00 ± 0.00 a	0.00 ± 0.00 a	5.02 ± 0.27 b
6	Heptane	0.00 ± 0.00 a	0.00 ± 0.00 a	16.07 ± 2.56 b
7	Dodecane	22.15 ± 0.81 c	0.00 ± 0.00 a	2.56 ± 0.26 b
8	Octane	0.00 ± 0.00 a	28.35 ± 0.66 b	31.38 ± 1.41 c
9	Benzene, propyl-	1.36 ± 0.19 b	0.00 ± 0.00 a	2.71 ± 0.92 b
10	Methyl tetradecanoate	6.71 ± 0.20 a	18.95 ± 0.49 c	16.74 ± 1.17 b
11	Pentadecanoic acid, methyl ester	0.00 ± 0.00 a	22.91 ± 2.28 b	19.79 ± 0.72 b
12	D-Limonene	0.00 ± 0.00 a	0.00 ± 0.00 a	0.54 ± 0.03 b
13	Butylated Hydroxytoluene	1.25 ± 0.02 b	0.00 ± 0.00 a	8.11 ± 0.21 c
14	Naphthalene, 1,2,3,4-tetrahydro-1, 1,1,6-trimethyl-	0.00 ± 0.00 a	6.84 ± 0.15 b	10.71 ± 1.68 c
Total		31.47	77.05	135.75

Values are mean ± standard deviation (SD, *n* = 3). The letters in the same row indicate that there is a significant difference between the groups (*p* < 0.05).

**Table 3 foods-12-02504-t003:** Changes in the concentration of volatile compounds produced from C18:0.

Number	Compounds	Concentration of Volatile Compounds (μg/L)		
		Blank	C16:0	C18:0 + LOX
1	Heptanal	0.00 ± 0.00 a	13.60 ± 1.52 b	20.73 ± 1.46 c
2	Nonanal	0.00 ± 0.00 a	0.00 ± 0.00 a	7.80 ± 0.40 b
3	2-Heptanone	0.00 ± 0.00 a	21.40 ± 0.78 b	25.46 ± 3.09 b
4	2-Nonanone	0.00 ± 0.00 a	0.00 ± 0.00 a	7.31 ± 0.28 b
5	Eicosane	0.00 ± 0.00 a	4.43 ± 0.32 b	0.00 ± 0.00 a
6	Heptane	0.00 ± 0.00 a	53.78 ± 4.18 c	46.48 ± 2.59 b
7	Octane	0.00 ± 0.00 a	37.23 ± 1.74 b	48.72 ± 1.58 c
8	Tridecane	0.00 ± 0.00 a	0.00 ± 0.00 a	0.56 ± 0.02 b
9	Benzene, propyl-	1.36 ± 0.19 b	0.00 ± 0.00 a	36.13 ± 1.82 b
10	Benzene, n-butyl-	12.34 ± 0.94 b	8.57 ± 0.24 a	9.45 ± 1.52 a
11	Toluene	0.00 ± 0.00 a	3.49 ± 0.26 b	7.28 ± 0.49 c
12	Ethylbenzene	0.00 ± 0.00 a	4.55 ± 0.22 b	7.33 ± 0.29 c
13	Dodecanoic acid, methyl ester	0.00 ± 0.00 a	19.17 ± 2.63 b	19.03 ± 1.15 b
14	2-Octanamine	0.00 ± 0.00 a	6.18 ± 0.51 b	11.22 ± 1.09 c
15	Butylated Hydroxytoluene	1.25 ± 0.02 a	0.00 ± 0.00 a	42.18 ± 1.93 b
16	Naphthalene	0.00 ± 0.00 a	0.00 ± 0.00 a	0.03 ± 0.00 b
Total		14.95	172.40	289.71

Values are mean ± standard deviation (SD, *n* = 3). The letters in the same row indicate that there is a significant difference between the groups (*p* < 0.05).

**Table 4 foods-12-02504-t004:** Changes in the concentration of volatile compounds produced from C18:1n9c.

Number	Compounds	Concentration of Volatile Compounds (μg/L)		
		Blank	C16:0	C18:1n9c + LOX
1	2-Decenal, (E)-	0.00 ± 0.00 a	40.38 ± 2.83 c	34.05 ± 1.32 b
2	Heptanal	0.00 ± 0.00 a	1.82 ± 0.21 b	12.41 ± 0.37 c
3	Decanal	0.00 ± 0.00 a	12.83 ± 0.46 b	19.03 ± 0.33 c
4	Hexanal	0.00 ± 0.00 a	18.77 ± 0.53 b	26.36 ± 1.77 c
5	Nonanal	0.00 ± 0.00 a	4.87 ± 0.06 b	8.52 ± 0.08 c
6	Octanal	0.00 ± 0.00 a	14.60 ± 0.57 b	16.98 ± 0.47 c
7	1-Nonanol	0.00 ± 0.00 a	43.14 ± 1.66 b	43.57 ± 2.71 b
8	2-Dodecenal	0.00 ± 0.00 a	0.00 ± 0.00 a	16.64 ± 0.82 b
9	1-Heptanol	0.00 ± 0.00 a	104.97 ± 2.53 b	129.74 ± 3.54 c
10	1-Octanol	0.00 ± 0.00 a	12.19 ± 0.13 b	13.14 ± 0.32 c
11	2-Decanone	0.00 ± 0.00 a	104.79 ± 2.42 b	112.02 ± 1.38 c
12	2-Nonanone	0.00 ± 0.00 a	47.18 ± 0.48 b	49.36 ± 0.57 c
13	2-Undecanone	0.00 ± 0.00 a	8.41 ± 0.26 b	10.12 ± 0.30 c
14	3-Octanone	0.00 ± 0.00 a	0.00 ± 0.00 a	20.83 ± 1.86 b
15	1-Heptene	0.00 ± 0.00 a	7.69 ± 0.46 b	11.10 ± 0.49 c
16	1-Octene	0.00 ± 0.00 a	10.80 ± 0.80 b	16.65 ± 1.08 c
17	Heptane	0.00 ± 0.00 a	0.00 ± 0.00 a	17.40 ± 2.91 b
18	Tetradecane	5.10 ± 0.08 b	4.30 ± 0.04 a	4.42 ± 0.04 a
19	Benzoic acid, ethyl ester	0.00 ± 0.00 a	14.05 ± 0.23 b	15.87 ± 1.04 c
20	Heptanoic acid, methyl ester	0.00 ± 0.00 a	113.10 ± 4.71 b	123.76 ± 7.27 b
21	Decanoic acid, methyl ester	0.00 ± 0.00 a	4.45 ± 0.14 b	5.68 ± 0.15 c
22	Hexanoic acid, methyl ester	0.00 ± 0.00 a	0.00 ± 0.00 a	2.19 ± 0.13 b
23	Formic acid, heptyl ester	0.00 ± 0.00 a	0.00 ± 0.00 a	91.12 ± 3.05 b
24	Formic acid, octyl ester	0.00 ± 0.00 a	0.00 ± 0.00 a	307.19 ± 8.30 b
25	Nonanoic acid, methyl ester	0.00 ± 0.00 a	120.26 ± 1.34 b	127.27 ± 0.53 c
26	Octanoic acid, methyl ester	0.00 ± 0.00 a	295.45 ± 1.06 b	294.10 ± 4.22 b
27	Octanoic acid, ethyl ester	0.00 ± 0.00 a	27.45 ± 0.71 b	30.14 ± 0.85 c
28	Butylated Hydroxytoluene	1.25 ± 0.02 b	0.00 ± 0.00 a	4.02 ± 0.12 c
Total		6.35	1020.50	1563.68

Values are mean ± standard deviation (SD, *n* = 3). The letters in the same row indicate that there is a significant difference between the groups (*p* < 0.05).

**Table 5 foods-12-02504-t005:** Changes in the concentration of volatile compounds produced from C18:2n6c.

Number	Compounds	Concentration of Volatile Compounds (μg/L)		
		Blank	C16:0	C18:2n6c + LOX
1	2,4-Nonadienal, (E,E)-	0.00 ± 0.00 a	9.54 ± 1.12 b	11.51 ± 2.21 b
2	2,4-Decadienal, (E,E)-	0.00 ± 0.00 a	11.72 ± 1.49 b	9.88 ± 0.42 b
3	Heptanal	0.00 ± 0.00 a	0.00 ± 0.00 a	45.13 ± 4.84 b
4	Hexanal	0.00 ± 0.00 a	21.95 ± 1.36 b	35.72 ± 1.43 c
5	Nonanal	0.00 ± 0.00 a	18.75 ± 1.52 b	32.04 ± 3.54 c
6	Heptanal	0.00 ± 0.00 a	27.94 ± 1.99 b	32.41 ± 1.80 c
7	2-Heptanone	0.00 ± 0.00 a	44.29 ± 1.71 b	57.96 ± 5.78 c
8	1-Butanone, 1-phenyl-	0.00 ± 0.00 a	15.66 ± 0.76 b	16.71 ± 1.68 b
9	1-Pentanone, 1-phenyl-	0.00 ± 0.00 a	16.90 ± 1.29 b	18.28 ± 1.21 b
10	3-Nonenoic acid, methyl ester	0.00 ± 0.00 a	11.56 ± 0.57 b	11.40 ± 0.69 b
11	Propane, 1,3-diiodo-	0.00 ± 0.00 a	0.00 ± 0.00 a	10.56 ± 1.02 b
12	Benzene, hexyl-	0.00 ± 0.00 a	10.04 ± 1.49 b	9.59 ± 0.83 b
13	Benzene, (2-methyl-1-propenyl)-	0.00 ± 0.00 a	0.00 ± 0.00 a	20.51 ± 1.31 b
14	3,4-Octadiene, 7-methyl-	0.00 ± 0.00 a	0.00 ± 0.00 a	7.29 ± 0.06 b
15	Benzene, 1-methyl-3-phenoxy-	0.00 ± 0.00 a	4.28 ± 0.30 b	4.10 ± 0.31 b
16	Hex-1-enylbenzene	0.00 ± 0.00 a	8.48 ± 0.67 b	8.54 ± 0.51 b
17	trans-1-Phenyl-1-pentene	0.00 ± 0.00 a	24.22 ± 4.07 c	12.20 ± 1.16 b
18	Benzene, 1-butenyl-, (E)-	0.00 ± 0.00 a	21.49 ± 1.50 b	25.58 ± 0.53 c
19	Undecanoic acid, methyl ester	0.00 ± 0.00 a	1.34 ± 0.10 b	0.00 ± 0.00 a
20	Hexane	0.00 ± 0.00 a	25.59 ± 3.29 b	29.25 ± 2.97 b
21	Benzene, propyl-	0.00 ± 0.00 a	103.56 ± 1.87 b	104.14 ± 24.79 b
22	Benzene, butyl-	0.00 ± 0.00 a	142.03 ± 16.54 b	151.74 ± 23.26 b
23	1-Hexanone, 1-phenyl-	0.00 ± 0.00 a	12.64 ± 1.89 b	10.64 ± 0.63 b
24	Benzene, pentyl-	0.00 ± 0.00 a	60.71 ± 9.13 b	60.28 ± 7.18 b
25	Nonanoic acid, 9-oxo-, methyl ester	0.00 ± 0.00 a	28.34 ± 7.52 b	0.00 ± 0.00 a
26	Methyl 8-(2-furyl) octanoate	0.00 ± 0.00 a	0.00 ± 0.00 a	1.38 ± 0.10 b
27	2-Heptenoic acid, methyl ester	0.00 ± 0.00 a	2.30 ± 0.10 b	0.00 ± 0.00 a
28	Benzenebutanoic acid, methyl ester	0.00 ± 0.00 a	2.77 ± 0.43 b	2.93 ± 0.06 b
29	6-Nonenoic acid, methyl ester	0.00 ± 0.00 a	17.55 ± 2.01 b	0.00 ± 0.00 a
30	7,10-Octadecadienoic acid, methyl ester	0.00 ± 0.00 a	26.53 ± 3.03 b	21.76 ± 2.37 b
31	8,11-Octadecadienoic acid, methyl ester	0.00 ± 0.00 a	819.49 ± 33.99 b	891.94 ± 81.73 b
32	9,11-Octadecadienoic acid, methyl ester, (E,E)-	0.00 ± 0.00 a	27.65 ± 1.47 b	0.00 ± 0.00 a
33	Nonanoic acid, 9-oxo-, methyl ester	0.00 ± 0.00 a	0.00 ± 0.00 a	36.41 ± 3.08 b
34	10-Undecenoic acid, methyl ester	0.00 ± 0.00 a	14.56 ± 1.28 b	13.55 ± 0.63 b
35	Heptanoic acid, methyl ester	0.00 ± 0.00 a	256.79 ± 18.02 b	0.00 ± 0.00 a
36	Decanoic acid, methyl ester	0.00 ± 0.00 a	7.14 ± 0.30 b	36.28 ± 0.94 c
37	Hexanoic acid, methyl ester	0.00 ± 0.00 a	125.36 ± 3.80 b	128.35 ± 1.96 b
38	Nonanoic acid, methyl ester	0.00 ± 0.00 a	39.41 ± 1.31 b	41.12 ± 2.68 b
39	9,12-Octadecadienoic acid, methyl ester, (E,E)-	0.00 ± 0.00 a	28.09 ± 2.09 b	32.94 ± 4.59 b
40	Methyl tetradecanoate	6.71 ± 0.20 b	0.00 ± 0.00 a	13.76 ± 2.28 c
41	Tetradecane	5.10 ± 0.08 a	4.29 ± 0.21 a	4.45 ± 0.87 a
42	Octanoic acid, methyl ester	0.00 ± 0.00 a	327.76 ± 6.89 b	380.37 ± 30.00 c
43	9,12-Octadecadienoic acid (Z,Z)-, methyl ester	0.00 ± 0.00 a	0.00 ± 0.00 a	27.68 ± 2.65 b
44	Dodecanoic acid, methyl ester	0.00 ± 0.00 a	2.71 ± 0.29 a	83.34 ± 14.25 b
45	Hexadecanoic acid, methyl ester	0.00 ± 0.00 a	8.39 ± 0.65 b	10.10 ± 0.54 c
46	9-Hexadecenoic acid, methyl ester, (Z)-	0.00 ± 0.00 a	7.23 ± 0.28 b	6.99 ± 0.66 b
47	Butylated Hydroxytoluene	1.25 ± 0.02 a	0.00 ± 0.00 a	9.86 ± 1.77 b
48	Naphthalene	0.00 ± 0.00 a	0.00 ± 0.00 a	1.14 ± 0.17 b
49	Oxime-, methoxy-phenyl-	0.00 ± 0.00 a	23.15 ± 2.15 b	0.00 ± 0.00 a
Total		13.06	2362.20	2469.81

Values are mean ± standard deviation (SD, *n* = 3). The letters in the same row indicate that there is a significant difference between the groups (*p* < 0.05).

**Table 6 foods-12-02504-t006:** Changes in the concentration of volatile compounds produced from C20:4.

Number	Compounds	Concentration of Volatile Compounds (μg/L)		
		Blank	C16:0	C20:4 + LOX
1	2,4-Decadienal, (E,E)-	0.00 ± 0.00	0.00 ± 0.00 a	3.21 ± 0.19 a
2	Hexanal	0.00 ± 0.00 a	2.44 ± 0.20 b	3.88 ± 0.35 c
3	Heptanal	0.00 ± 0.00 a	0.00 ± 0.00 a	15.17 ± 2.41 b
4	Benzene, decyl-	0.00 ± 0.00 a	0.38 ± 0.37 a	0.00 ± 0.00 a
5	5,8,11,14-Eicosatetraenoic acid, ethyl ester, (all-Z)-	0.00 ± 0.00 a	0.00 ± 0.00 a	22.65 ± 2.26 b
6	8-Heptadecene	0.00 ± 0.00 a	0.61 ± 0.02 b	1.37 ± 0.17 c
7	Dodecane	22.15 ± 0.81 a	135.95 ± 11.07 b	229.05 ± 23.00 c
8	Tetradecane	5.10 ± 0.08 b	3.25 ± 0.13 b	5.64 ± 1.09 c
9	Undecane	5.10 ± 0.07 a	469.21 ± 14.74 b	769.46 ± 30.64 c
10	Heptylcyclohexane	14.77 ± 10.48 a	0.00 ± 0.00 a	10.04 ± 0.21 a
11	Cyclohexane, pentyl-	0.00 ± 0.00 a	4.46 ± 0.29 b	0.00 ± 0.00 a
12	Benzene, 1,2,4-trimethyl-	0.00 ± 0.00 a	72.94 ± 6.27 b	73.54 ± 5.75 b
13	Benzene, propyl-	0.00 ± 0.00 a	20.24 ± 1.31 b	27.20 ± 2.88 c
14	p-Xylene	0.00 ± 0.00 a	0.00 ± 0.00 a	6.42 ± 0.13 b
15	Toluene	0.00 ± 0.00 a	2.40 ± 0.24 b	4.51 ± 0.15 c
16	Benzene, 1,3-dimethyl-	0.00 ± 0.00 a	18.79 ± 1.20 b	30.59 ± 0.56 c
17	Benzene, 1,3,5-trimethyl-	0.00 ± 0.00 a	13.62 ± 0.97 b	13.12 ± 1.53 b
18	o-Xylene	0.00 ± 0.00 a	8.32 ± 0.28 b	25.80 ± 5.30 c
19	Benzene, 1-ethyl-2-methyl-	0.00 ± 0.00 a	3.34 ± 0.07 b	6.54 ± 0.99 c
20	Ethylbenzene	0.00 ± 0.00 a	0.90 ± 0.03 b	1.20 ± 0.26 b
21	Decanoic acid, methyl ester	0.00 ± 0.00 a	0.00 ± 0.00 a	1.69 ± 0.28 a
22	Dodecanoic acid, methyl ester	0.00 ± 0.00 a	0.00 ± 0.00 a	3.47 ± 0.31 a
23	n-Hexadecanoic acid	3.41 ± 2.41 a	0.00 ± 0.00 a	0.63 ± 0.08 a
24	D-Limonene	0.00 ± 0.00 a	141.47 ± 195.47 a	577.24 ± 15.00 b
25	Naphthalene, decahydro-, trans-	0.00 ± 0.00 a	21.02 ± 26.72 a	0.00 ± 0.00 a
26	cis-Decalin, 2-syn-methyl-	0.00 ± 0.00 a	0.00 ± 0.00 a	268.51 ± 33.67 b
Total		50.53	919.34	2100.93

Values are mean ± standard deviation (SD, *n* = 3). The letters in the same row indicate that there is a significant difference between the groups (*p* < 0.05).

**Table 7 foods-12-02504-t007:** Changes in the concentrations of volatile compounds produced from C20:5.

Number	Compounds	Concentration of Volatile Compounds (μg/L)		
		Blank	C16:0	C20:5 + LOX
1	Quinoline, 1,2-dihydro -2,2,4-trimethyl-	0.00 ± 0.00 a	0.00 ± 0.00 a	1.28 ± 0.24 b
2	Decane, 2-methyl-	0.00 ± 0.00 a	0.00 ± 0.00 a	4.39 ± 0.32 b
3	Decane, 4-methyl-	0.00 ± 0.00 a	0.00 ± 0.00 a	13.76 ± 1.03 b
4	Cyclohexane, butyl-	0.00 ± 0.00 a	0.00 ± 0.00 a	24.07 ± 1.21 b
5	Dodecane	22.15 ± 0.81 c	6.72 ± 0.40 a	9.83 ± 0.40 b
6	Tetradecane	5.10 ± 0.08 c	1.90 ± 0.02 a	2.35 ± 0.19 b
7	Undecane	5.10 ± 0.07 a	59.16 ± 2.46 b	81.16 ± 11.25 c
8	Undecane, 2,6-dimethyl-	0.00 ± 0.00 a	0.00 ± 0.00 a	4.53 ± 0.23 b
9	Cyclohexane, pentyl-	0.00 ± 0.00 a	0.00 ± 0.00 a	9.71 ± 1.08 b
10	Benzene, 1,2,4,5-tetramethyl-	0.00 ± 0.00 a	0.00 ± 0.00 a	5.71 ± 0.24 b
11	p-Xylene	0.00 ± 0.00 a	11.67 ± 0.73 b	20.56 ± 1.63 c
12	Toluene	0.00 ± 0.00 a	0.00 ± 0.00 a	15.45 ± 2.65 b
13	Mesitylene	0.00 ± 0.00 a	0.00 ± 0.00 a	23.41 ± 0.58 b
14	o-Xylene	0.00 ± 0.00 a	179.16 ± 5.23 c	161.18 ± 6.58 b
15	Benzene, 1-ethyl-2-methyl-	0.00 ± 0.00 a	0.00 ± 0.00 a	17.02 ± 2.32 b
16	Benzene, 1-methyl-2-(1-methylethyl)-	0.00 ± 0.00 a	4.39 ± 0.24 b	0.00 ± 0.00 a
17	Ethylbenzene	0.00 ± 0.00 a	11.67 ± 1.07 b	14.78 ± 0.20 c
18	Benzene, (1-methylethyl)-	0.00 ± 0.00 a	0.00 ± 0.00 a	2.68 ± 0.39 b
19	5,8,11,14-Eicosatetraenoic acid,ethyl ester, (all-Z)-	0.00 ± 0.00 a	6.52 ± 0.10 b	0.00 ± 0.00 a
20	Decanoic acid, methyl ester	0.00 ± 0.00 a	45.29 ± 1.01 b	50.07 ± 1.09 c
21	Octanoic acid, methyl ester	0.00 ± 0.00 a	16.71 ± 0.43 a	17.55 ± 1.95 a
22	Dodecanoic acid, methyl ester	0.00 ± 0.00 a	20.79 ± 1.09 b	25.72 ± 1.23 c
23	2-(6-Ethoxy-4-methyl-quinazolin-2-ylamino) -5-methyl-pyrimidine-4, 6-diol	0.00 ± 0.00 a	0.00 ± 0.00 a	0.31 ± 0.01 b
24	Naphthalene, 1-methyl-	0.00 ± 0.00 a	0.60 ± 0.01 c	0.27 ± 0.02 b
25	trans-Decalin, 2-methyl-	0.00 ± 0.00 a	0.00 ± 0.00 a	20.23 ± 0.36 b
26	Tridecanoic acid, 12-methyl-,methyl ester	0.00 ± 0.00 a	0.00 ± 0.00 a	0.77 ± 0.00 b
27	Naphthalene, decahydro-2-methyl-	0.00 ± 0.00 a	0.00 ± 0.00 a	14.43 ± 0.65 b
28	Naphthalene, decahydro-, trans-	0.00 ± 0.00 a	27.78 ± 1.34 b	26.31 ± 0.87 b
Total		32.35	392.36	567.53

Values are mean ± standard deviation (SD, *n* = 3). The letters in the same row indicate that there is a significant difference between the groups (*p* < 0.05).

**Table 8 foods-12-02504-t008:** Changes in the concentrations of volatile compounds produced from C22:6n3.

Number	Compounds	Volatile Compounds Concentration (μg/L)		
		Blank	C16:0	C22:6n3 + LOX
1	Quinoline, 1,2-dihydro-2,2,4-trimethyl-	0.00 ± 0.00 a	0.00 ± 0.00 a	15.01 ± 1.44 b
2	Cyclohexane, butyl-	0.00 ± 0.00 a	22.42 ± 0.83 b	24.78 ± 1.73 b
3	trans-Decalin, 2-methyl-	0.00 ± 0.00 a	0.00 ± 0.00 a	10.36 ± 1.14 b
4	Decane	0.00 ± 0.00 a	340.40 ± 25.88 b	293.78 ± 28.78 b
5	Dodecane	22.15 ± 0.81 a	23.68 ± 0.97 ab	29.24 ± 4.16 b
6	Tetradecane	5.10 ± 0.08 b	5.30 ± 0.87 b	2.88 ± 0.62 a
7	Undecane	5.10 ± 0.07 a	296.20 ± 35.44 b	336.59 ± 37.62 b
8	Undecane, 2,6-dimethyl-	0.00 ± 0.00 a	0.00 ± 0.00 a	4.94 ± 1.08 b
9	Tridecane	1.12 ± 0.07 a	3.67 ± 0.66 c	2.72 ± 0.08 b
10	Cyclohexane, pentyl-	8.95 ± 0.10 a	27.61 ± 2.57 b	28.22 ± 3.51 b
11	Benzene, 1,2,3,4-tetramethyl-	0.00 ± 0.00 a	6.80 ± 1.43 b	16.68 ± 2.90 c
12	Benzene, 1-ethyl-2,3-dimethyl-	0.00 ± 0.00 a	0.00 ± 0.00 a	24.93 ± 2.54 b
13	Benzene, 1-methyl-2-(1-methylethyl)-	0.00 ± 0.00 a	0.00 ± 0.00 a	24.63 ± 0.89 b
14	Benzene, 2-ethyl-1,4-dimethyl-	0.00 ± 0.00 a	0.00 ± 0.00 a	38.45 ± 1.81 b
15	p-Xylene	0.00 ± 0.00 a	85.43 ± 4.14 c	77.14 ± 2.69 b
16	Toluene	0.00 ± 0.00 a	0.00 ± 0.00 a	11.79 ± 1.04 b
17	Benzene, 1,3-dimethyl-	0.00 ± 0.00 a	92.40 ± 2.31 c	78.76 ± 2.73 b
18	Mesitylene	0.00 ± 0.00 a	0.00 ± 0.00 a	16.28 ± 3.59 b
19	Benzene, 1,2,3-trimethyl-	0.00 ± 0.00 a	0.00 ± 0.00 a	9.66 ± 1.87 b
20	o-Xylene	0.00 ± 0.00 a	221.15 ± 6.12 c	100.60 ± 5.17 b
21	Ethylbenzene	0.00 ± 0.00 a	95.34 ± 7.28 c	81.36 ± 3.70 b
22	Benzene, 1,2,3,5-tetramethyl-	0.00 ± 0.00 a	29.43 ± 3.19 b	39.19 ± 2.58 c
23	Decanoic acid, methyl ester	0.00 ± 0.00 a	248.19 ± 7.81 b	286.13 ± 18.01 c
24	Methyl tetradecanoate	0.00 ± 0.00 a	0.00 ± 0.00 a	32.14 ± 3.12 b
25	Undecanoic acid, methyl ester	0.00 ± 0.00 a	8.36 ± 1.42 b	7.99 ± 1.16 b
26	Octanoic acid, methyl ester	0.00 ± 0.00 a	0.00 ± 0.00 a	105.71 ± 1.70 b
27	Dodecanoic acid, methyl ester	0.00 ± 0.00 a	109.95 ± 2.40 b	207.83 ± 38.88 c
28	Hexadecanoic acid, methyl ester	0.00 ± 0.00 a	0.00 ± 0.00 a	2.04 ± 0.16 b
29	Benzocycloheptatriene	0.00 ± 0.00 a	3.08 ± 0.15 b	2.95 ± 0.05 b
30	Butylated Hydroxytoluene	1.25 ± 0.02 a	0.00 ± 0.00 a	7.03 ± 1.04 b
31	Naphthalene	0.00 ± 0.00 a	23.34 ± 1.34 b	23.64 ± 1.77 b
Total		43.67	1642.75	1943.45

Values are mean ± standard deviation (SD, *n* = 3). The letters in the same row indicate that there is a significant difference between the groups (*p* < 0.05).

## Data Availability

The data presented in this study are available on request from the corresponding author.

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
