# Peer review of "Formation of Lipid-Derived Flavors in Dry-Cured Mackerel (Scomberomorus niphonius) via Simulation of Autoxidation and Lipoxygenase-Induced Fatty Acid Oxidation"

_foods, 2023, doi:10.3390/foods12132504_

Round 1
Reviewer 1 Report
You compare your results with those from different STUDY, BUT same methods. Since, mackerel have been investigated with different seafood processing methods which useful comparison with many of the previously published studies (https://doi.org/10.1016/j.foodchem.2023.135559, https://doi.org/10.1111/jfpp.14059 etc)
The researchers should be explained the validation methods in analyzes
The authors sholud be add some information results of analyzes in conclusion section .
Most of Figures are of poor quality (at least in the the PDF I reviewed). I suggest authors make sure they are sharp and of high resolution for publication.
Reviewer 2 Report
The objective of the manuscript was of the purification and properties (storage time, temperature and pH) of pipoxygenase enzyme from dry-cured mackere and explore the main metabolic pathways of key flavors in dry-cured mackerel with different substrates (of palmitic acid (C16:0), stearic acid (C18:0), oleic acid (C18:2n9c), linoleic acid (C18:2n6c), arachidonic acid (C20:4), EPA (C20:5), DHA (C22:6n3)).
The manuscript in generally is good and interesting. The introduction has the information necessary. The methodology is accord with the experiment. The conclusion is clary and concision. The results and discussion is good, but in results I have one question. ¿Why in figure 2, they are only three substrates and they are not seven substrates.
Reviewer 3 Report
Manuscript ID: foods-2461363
Title: “Formation of lipid-derived flavors in dry-cured mackerel (Scomberomorus niphonius) by simulating fatty acids oxidation”
General comments
In this paper, the lipoxygenase extracted from dry-cured mackerel was purified and the autoxidation and LOX-induced oxidation of palmitic acid, stearic acid, oleic acid, linoleic acid, arachidonic acid, EPA, and DHA were simulated.
The basic idea of the manuscript is good, and it could be of practical interest.
The manuscript is generally well written with a logic structure.
Applied methods can be considered as adequate to investigate the research questions.
However, the authors have published a similar article in which there are paragraphs that are repeated in this: “Liu, Qiaoyu and Lei, Menglin and Lin, Jianjun and Zhao, Wenhong and Zeng, Xiaofang and Bai, Weidong, The Role of Lipoxygenases and Autoxidation in the Dry-Cured Processing of Mackerel (Scomberomorus niphonius). Available at SSRN: https://ssrn.com/abstract=4241946”
I think that the title should include the word “lipoxygenase”.
I think the introduction does not summarize well the theoretical background of the research.
The first paragraph of the introduction corresponds to the guide for the author.
Separate the brackets from the references in the text from the words.
Specify how many times the experiments were repeated number of replicates in each case.
Was the fish chilled or frozen?
In the phrase: "The edible salt (4% of dry fish weight) was evenly spread on both sides of fish in the corresponding group”, what groups are you referring to? Was there a control group?
Why are the tables numbered 2A, 2B, ... and not 2, 3, 4,…?
The letters of the statistics should be explained at the table footnotes.
Round 2
Reviewer 3 Report
Many of the comments have been emended in the revised manuscript. The revised manuscript is much improved;